# Determinants of Exclusive Breastfeeding Cessation in the Early Postnatal Period among Culturally and Linguistically Diverse (CALD) Australian Mothers

**DOI:** 10.3390/nu11071611

**Published:** 2019-07-16

**Authors:** Felix Akpojene Ogbo, Osita Kingsley Ezeh, Sarah Khanlari, Sabrina Naz, Praween Senanayake, Kedir Y. Ahmed, Anne McKenzie, Olayide Ogunsiji, Kingsley Agho, Andrew Page, Jane Ussher, Janette Perz, Bryanne Barnett AM, John Eastwood

**Affiliations:** 1Translational Health Research Institute, School of Medicine, Western Sydney University, Campbelltown Campus, Locked Bag 1797, Penrith, NSW 2571, Australia; 2General Practice Unit, Prescot Specialist Medical Centre, Welfare Quarters, Makurdi, Benue State 972261, Nigeria; 3School of Science and Health, Western Sydney University, Campbelltown Campus, Locked Bag 1797, Penrith, NSW 2571, Australia; 4Department of Community Paediatrics, Sydney Local Health District, Croydon Community Health Centre, 24 Liverpool Street, Croydon, NSW 2132, Australia; 5School of Medicine and Public Health, University of Newcastle, University Drive, Callaghan, NSW 2308, Australia; 6Child and Family Health Nursing, Primary & Community Health, South Western Sydney Local Health District, Narellan, NSW 2567, Australia; 7School of Nursing and Midwifery, Western Sydney University, Liverpool Campus, Locked Bag 1797, Penrith, NSW 2571, Australia; 8St John of God Raphael Services, Blacktown, NSW 2148, Australia; 9Ingham Institute for Applied Medical Research, 1 Campbell Street, Liverpool, NSW 2170, Australia; 10School of Women’s and Children’s Health, The University of New South Wales, Kensington, Sydney, NSW 2052, Australia; 11Menzies Centre for Health Policy, Charles Perkins Centre, School of Public Health, Sydney University, Sydney, NSW 2006, Australia; 12School of Public Health, Griffith University, Gold Coast, QLD 4222, Australia; 13Sydney Institute for Women, Children and their Families, Sydney Local Health District, Camperdown, NSW 2050, Australia

**Keywords:** exclusive breastfeeding, Australia, skin-to-skin, culturally and linguistically diverse (CALD)

## Abstract

There are limited epidemiological data on exclusive breastfeeding (EBF) among culturally and linguistically diverse (CALD) Australian mothers to advocate for targeted and/or culturally-appropriate interventions. This study investigated the determinants of EBF cessation in the early postnatal period among CALD Australian mothers in Sydney, Australia. The study used linked maternal and child health data from two local health districts in Australia (*N* = 25,407). Prevalence of maternal breastfeeding intention, skin-to-skin contact, EBF at birth, discharge, and the early postnatal period (1–4 weeks postnatal), were estimated. Multivariate logistic regression models were used to investigate determinants of EBF cessation in the early postnatal period. Most CALD Australian mothers had the intention to breastfeed (94.7%). Skin-to-skin contact (81.0%), EBF at delivery (91.0%), and at discharge (93.0%) were high. EBF remained high in the early postnatal period (91.4%). A lack of prenatal breastfeeding intention was the strongest determinant of EBF cessation (adjusted odds ratio [aOR] = 23.76, 95% CI: 18.63–30.30, for mothers with no prenatal breastfeeding intention and aOR = 6.15, 95% CI: 4.74–7.98, for those undecided). Other significant determinants of EBF cessation included a lack of partner support, antenatal and postnatal depression, intimate partner violence, low socioeconomic status, caesarean birth, and young maternal age (<20 years). Efforts to improve breastfeeding among women of CALD backgrounds in Australia should focus on women with vulnerabilities to maximise the benefits of EBF.

## 1. Introduction

Global health organisations (such as the World Health Organization and the United Nations Children’s Fund, WHO/UNICEF) recommend exclusive breastfeeding (EBF) for the first six months of life [1]. EBF is defined as providing the infant human breastmilk only, and when needed, oral rehydration solution, or drops/syrups of vitamins, minerals, or medicines [2]. Breast milk has a high proportion of fat, protein, sugar, and water that is required for infant growth and development compared to formula milk, as well as the immunologic substances to effectively protect against infectious diseases for the infant [3,4,5]. Appropriate EBF protects against childhood obesity [6] and has the potential to increase infant cognitive functioning [7]. EBF not only benefits the infant but is also associated with improved maternal health (e.g., a reduced risk developing of type 2 diabetes mellitus) [8] and improved household productivity due to no cost of human milk [9]. 

In Australia, culturally and linguistically diverse (CALD) is a term used for communities with diverse ethnic backgrounds, traditions, food, nationality, language, dress, societal structures, art, and religious characteristics [10]. While it may be debatable that such a ‘label’ exists to characterize such diverse populations, it is nevertheless one measure that can identify a sub-group in the Australian population who are often socioeconomically vulnerable, and for the purpose of focused research and programmatic interventions [11,12]. Past research suggests that CALD subgroups are more likely to experience adverse health outcomes, including depressive symptoms [13] and poor uptake of cancer screening [14], compared to the non-CALD populations. 

Many qualitative studies conducted in Australia have reported that most CALD women value optimal breastfeeding practices, but a lack of access to traditional post-birth practices [15], stigma, and shame around public breastfeeding and ambivalence towards breastfeeding support [16,17], were barriers to appropriate breastfeeding. However, there is limited epidemiological data on the determinants of inappropriate EBF among CALD subgroups to inform targeted interventions. Quantitative research in the general Australian population suggests that intimate partner violence [18], a lack of partner support [18], assisted delivery [18,19], low socioeconomic status [20], lower maternal age (<25 years) [19], a mother not having intention to breastfeeding [21,22,23], and self-reported depressive symptoms [21] were associated with cessation of EBF. Notably, it is unclear whether these factors are relevant to mothers from CALD backgrounds as culturally-appropriate and focused intervention strategies are potentially more effective and less costly than traditional interventional approaches [12].

Understanding the factors that influence a mother’s decision to initiate, cease, or continue EBF in the early postnatal period is essential as it can provide important information and opportunities for specific interventions. In addition to providing relevant data on EBF among CALD Australian mothers, this study seeks to provide an evidence base on EBF among CALD mothers for breastfeeding policy advocacy and/or evaluation of relevant social and health services for mothers of CALD backgrounds in Australia. This study aimed to investigate the determinants of EBF cessation in the early postnatal period among CALD Australian mothers in Sydney, New South Wales.

## 2. Materials and Methods 

### 2.1. Data Source

The study was conducted based on retrospective maternal and child health data of all live births in public health facilities in Sydney Local Health District (SLHD) and South Western Sydney Local Health District (SWSLHD) between 2014 and 2016 (*N* = 25,407). These data were routinely collected as part of standard care provided during pregnancy and the postnatal period. Antenatal information that included socio-demographic characteristics, history of any previous pregnancy, probable depression based on the Edinburgh Postnatal Depression Scale (EPDS), and mothers’ breastfeeding intention were collected by qualified midwives at the first prenatal care visit. Birth and postnatal data such as information on skin-to-skin contact, EBF at discharge and postnatally, were also obtained immediately after birth and during postnatal visits by qualified child and family health nurses (CFHN). During the first prenatal visit, women were asked to identify whether they belong to CALD, non-CALD, or Aboriginal or Torres Strait Islander subpopulations, and this information was entered into the database. CALD population was defined based on the Australian Bureau of Statistics description for the subgroup [10]. These maternal child health (MCH) data were stored in the local health district’s Information Management & Technology Division (IM&TD) database. We obtained the perinatal data from the IM&TD, which were cleaned and linked using individual identifiers, and coded for analysis. 

### 2.2. Study Setting

In Sydney, the SLHD and SWSLHD cover 52% of the metropolitan area, with an estimated population of 1.6 million people of different cultural backgrounds [24,25]. SLHD is located in the centre and inner west of Sydney, while SWSLHD is located in the south-western region of Sydney. A number of maternal and child health services are provided to all communities across both districts, including the most socioeconomically disadvantaged populations. 

### 2.3. Outcome Variables

The main outcome variable was EBF in the early postnatal period (defined as 1–4 weeks post-birth). EBF was measured using the National Health and Medical Research Council (NHMRC) infant feeding guidelines [26], consistent with the WHO/UNICEF definitions for assessing infant and young child feeding practices [1]. EBF was defined as the proportion of infants who received only breast milk (including expressed milk) but allowed oral rehydration solution, syrups of vitamins/medicines. The prevalence of mothers’ breastfeeding intention, skin-to-skin contact, EBF at delivery and discharge by the determining factors were also measured in the study. 

EBF at delivery was defined as the proportion of infants who received only breast milk in the first 24 h post-birth, while EBF at discharge was measured as the proportion of infants who received only breast milk 24 h preceding discharge from the maternity unit. EBF in the early postnatal visit (1–4 weeks postnatal) referred to the proportion of infants who received only breast milk 24 h prior to this postnatal health visit by the CFHN. To assess mothers’ breastfeeding intention, they were asked the following question: “Do you plan to breastfeed your child?” The study also considered skin-to-skin contact as past studies have indicated that skin-to-skin contact is the most effective strategy to promote, protect, and support EBF in early life, irrespective of the mode of birthing (vaginal or caesarean births) [27,28,29]. Skin-to-skin contact (SSC) was defined as placing the naked baby on the mother’s bare chest or abdomen immediately or less than 10 min after birth or soon afterwards [30]. 

In each local health district, assessment of both the mother and baby is conducted between the first and fourth week post-birth by a CFHN during a universal health home visit. Relevant health information (including anthropometric measurement of the baby and assessment of infant feeding practices of the mother) are obtained and entered into the IM&TD database.

### 2.4. Study Factors

The exposure variables were broadly categorised into socio-demographic and health factors, which were selected based on previous studies [18,19,22,23] and data availability. Socio-demographic factors included maternal age (categorised as <20 years, 20–34 years or ≥35 years), socioeconomic status (SES, categorised as high, middle or low), maternal cigarette smoking in pregnancy (categorised as yes or no), partner support (categorised as yes, not sure or no), and major nationality groups (categorised as Oceania, North-West Europe, Southern-Eastern Europe, North Africa and The Middle East, South-East Asia, North-East Asia, Southern and Central Asia, Americas or Sub-Saharan Africa).

Health factors included pre-existing maternal health problems (such as diabetes mellitus and/or hypertension, categorised as yes or no), history of intimate partner violence (IPV, categorised as yes or no), type of delivery (categorised as normal vaginal, assisted vaginal or caesarean), and self-reported antenatal and postnatal depressive symptoms (categorised as score ≥13 or score <13 on the EPDS) [31]. Maternal breastfeeding intention was also considered a potential factor for the cessation of EBF in the early post-birth period based on past studies [21,22,23]. 

SES was calculated using the Socio-Economic Index for Areas (SEIFA). SEIFA is an indicator created by the Australian Bureau of Statistics using principal component analysis. It ranks areas (including a mother’s address provided) in Australia according to relative socio-economic advantage and disadvantage [32]. In this study, deciles of SES were categorised into high (top 10% of the population), middle (middle 80%), and low (bottom 10%) groups, in line with previously published studies [13,31]. In accordance with NSW Health policy [33], IPV information was collected from mothers based on the following questions: (i) “within the last year have you been hit, slapped or hurt in other ways by your partner or ex-partner?”, physical IPV; and (ii) “are you frightened of your partner or ex-partner?”, psychological IPV.

### 2.5. Statistical Analysis

The analytical approach followed previously published studies [18,31]. Briefly, preliminary analyses were conducted to calculate frequencies of the study outcomes (i.e., breastfeeding intention, skin-to-skin contact, and EBF at delivery, discharge and postnatally) and cross-tabulations with study factors. This was followed by univariate regression models to examine the association between each study factor and cessation of EBF in the early postnatal period. Multivariate logistic regression analyses that adjusted for confounders was conducted to investigate the potential study factors that were associated with cessation of EBF in the early postnatal period among CALD mothers. Models adjusted for the potential confounding factors of birthing facility, the gender of the baby, maternal alcohol intake, and maternal body mass index, as well as socio-economic and health factors [21,34]. Odds ratios, with 95% confidence intervals, were calculated as the measure of association between the risk factors and cessation of EBF.

Our study also investigated the potential effect of missing data on the estimated odds ratios in sensitivity analyses that used an imputed dataset, based on the original data, which comprised complete information for EBF in the early postnatal period. Multiple imputations by chained equations were employed, which assume that data were missing at random [35]. This analytical approach also assumes that the known characteristics of study respondents can be used to examine the characteristics of participants with missing data [36]. All study factors and the outcome variable in the main analysis were included in the multiple imputation models. Revised odds ratios from the imputed data were generated using the *mim* command, for comparison with the complete case analyses. Sensitivity analyses were conducted based on 25 multiple imputations [37], and all analyses were conducted in Stata (Stata Corp, version 15.0, College Station, TX, USA). 

### 2.6. Ethics

The Sydney Local Health District and South Western Sydney Local Health District Human Research Ethics Committees approved the collection of the data from the IM&TD database and subsequent analysis. Approval numbers HREC: LNR/11/LPOOL/463; SSA: LNRSSA/11/LPOOL/464 and Project No: 11/276 LNR; Protocol No X12-0164 and LNR/12/RPAH/266. 

## 3. Results

### 3.1. Characteristics of the Study Population

The majority of mothers were from South-East Asia (24.5%) and North Africa and The Middle East (23.0%), while the lowest proportion of mothers were from North-West Europe (1.5%) (Table 1).

### 3.2. Breastfeeding Patterns by Study Factors

The proportion of CALD Australian mothers who exclusively breastfed in the early postnatal period varied by global regions, with mothers from South-East Asia (26.9%), North Africa and the Middle East (33.8%), and Oceania (12.2%) representing the highest percentage (Table 2). Mothers from the high SES category had a higher proportion of EBF (63.1%) compared to those from middle and low SES categories (33.4% and 3.5%, respectively). 

In the antenatal period, almost all mothers intended to breastfeed their babies (94.7%). Approximately 81% of mothers practised skin-to-skin contact, while 91% and 93% exclusively breastfed at delivery and discharge, respectively. A sub-analysis of the data (i.e., the estimation of the confidence interval and *p*-value around the estimates, data not shown) suggested that there was no significant difference between EBF prevalence at delivery and discharge. In the early postnatal period, EBF remained high among CALD Australian mothers (91.4%; Table 2).

### 3.3. Determinants of Exclusive Breastfeeding Cessation in the Early Postnatal Period

The study showed that mothers who indicated no prenatal breastfeeding intention or who were undecided about breastfeeding during pregnancy were more likely to cease EBF in the early postnatal period compared to those who indicated prenatal breastfeeding intention in the complete case analyses (adjusted odds ratio (aOR) = 23.76, 95% CI 18.63–30.30, *p* < 0.001, for those with no prenatal breastfeeding intention and aOR = 6.15, 95% CI 4.74–7.98, *p* < 0.001, for those undecided) (Table 3). 

Mothers who reported not having a supportive partner during pregnancy were more likely to stop EBF in the early post-birth period compared to those who reported receiving support from their partner during pregnancy (aOR = 1.69, 95% CI: 1.20–2.38, *p* = 0.003). Mothers from higher SES groups were less likely to discontinue EBF in the early postnatal period compared to those from lower SES groups (aOR = 0.48, 95% CI: 0.32–0.71, *p* < 0.001 for high SES and aOR = 0.85, 95% CI: 0.85–0.99, *p* = 0.044 for middle). Low maternal age (<20 years) was associated with cessation of EBF in the early postnatal period compared to middle reproductive-aged mothers (20–34 years; aOR = 1.72, 95% CI: 1.37–2.15, *p* < 0.001). Mothers who reported tobacco smoking during pregnancy were more likely to stop EBF in the early postnatal period compared to their counterparts who reported not smoking (aOR = 3.39, 95% CI: 2.56–4.49, *p* < 0.001) (Table 3).

The odds of stopping EBF in the immediate postpartum period were higher among mothers who reported antenatal depressive symptoms (EPDS ≥ 13; aOR = 1.50, 95% CI: 1.20–1.89, *p* < 0.001), and those who reported postnatal depressive symptoms (EPDS ≥ 13; aOR = 2.07, 95% CI: 1.55–2.77, *p* < 0.001). The likelihood of discontinuing EBF in the immediate postnatal period was higher among mothers who reported a history of psychological intimate partner violence (aOR = 1.66, 95% CI: 1.10–2.53, *p* = 0.017), and those who had caesarean births (aOR = 1.35, 95% CI: 1.18–1.55, *p* < 0.001). The odds of ceasing EBF in the immediate postpartum period were lower among mothers from all major nationality groups compared to their counterparts in the reference group (Oceania) (Table 3). The effects of maternal age > 35 years, physical intimate partner violence, and having pre-existing maternal health problems on ceasing EBF in the immediate postnatal period were statistically significant in the imputation data compared to those in the complete case analysis. Furthermore, the 95% CI of the aOR for no maternal breastfeeding intention and major nationality group based on the imputation data is much narrower than the one obtained based on the original data set, possibly reflecting the effect of missing data or small sample size (Table 3). 

## 4. Discussion

Our study indicates that most CALD Australian mothers had the intention to breastfeed (94.7%). Eighty-one percent of mothers practised skin-to-skin contact, while 91% and 93% exclusively breastfed at delivery and discharge, respectively. Notably, EBF remained high in the early postnatal period among CALD Australian mothers (91.4%). A lack of maternal prenatal breastfeeding intention (no prenatal breastfeeding intention or undecided) was the strongest risk factor for the cessation of EBF in the early postpartum period. Other significant factors associated with the cessation of EBF in the early postnatal period included a lack of partner support, antenatal and postnatal depressive symptoms, psychosocial IPV, caesarean birthing, low socioeconomic status, and young maternal age (<20 years).

Consistent with past reports from Australia [21] and internationally [38,39,40], the present study indicates that a mother’s prenatal intention not to breastfeed or undecided were the strongest risk factors for the cessation of EBF in the early postnatal period among CALD Australian women. Studies suggest that a good personal attitude towards breastfeeding, encouraging social norms for breastfeeding at home and work and a personal dislike for formula feeding, were essential to a mother’s decision to have a definite intention to breastfeed [38,41]. In addition, the level of social support for the mother at the time of breastfeeding from the partner, grandmothers/in-laws [42], or peers [43,44], and the mother’s own attitude to breastfeeding were influential to a mother’s breastfeeding behaviour [38]. Our study also suggests that the lack of a supportive partner was associated with cessation of EBF in the early postnatal period. While studies have indicated that partners are willing to provide the required support to improve breastfeeding outcomes for both the mother and baby, a lack of appropriate and/or conflicting information from health practitioners to fathers have been flagged as constraints to fathers’ full participation in breastfeeding support [45,46]. The involvement of fathers (in addition to grandmothers/in-laws if present) in prenatal breastfeeding education sessions and postnatal support are key priority areas for improving breastfeeding among mothers [47]. 

In Australia, many epidemiological studies have indicated that higher maternal SES was associated with EBF [19,20]. Our study shows that CALD mothers from higher SES groups were less likely to cease EBF in the early postnatal period compared to those from lower SES groups. Similarly, evidence has shown that infants whose fathers were from high SES groups were more likely to be breastfed up to 6-weeks post-birth compared to infants with fathers of low SES [47]. Possible reasons for why higher SES mothers (CALD and non-CALD) engage in optimal breastfeeding may include increased uptake of breastfeeding-related information and better skills in negotiating flexible workplace hours, creating opportunities for breastfeeding [48,49]. Research shows that young maternal age (<20 years) was associated with increased risk of mental health issues, with subsequent impacts on employment and socioeconomic position in later life [50]. The present study shows that young maternal age (<20 years) is associated with cessation of EBF in the early postnatal period, in line with previously published studies [18,19]. In many Australian communities, charitable organisations provide parenting support, life support, and educational opportunities for young mothers and their babies to lead healthy and productive lives in their local communities [51]. Integration of these services with antenatal and postnatal breastfeeding services (where practicable) would be essential to improve breastfeeding outcomes among young mothers. 

Our study demonstrates an association between self-reported maternal antenatal and postnatal depressive symptoms (EPDS ≥ 13) and cessation of EBF within 4-weeks postnatally. Past studies have indicated that perinatal depression is associated with suboptimal breastfeeding, including cessation of EBF in the early postnatal period [52,53,54]. Our finding highlights the importance of routine perinatal depression screening in Australia [55]. Detailed information on policy and research implications of Australian perinatal depression screening has been published elsewhere [13,31,55,56]. Furthermore, the present study shows that self-reported psychological intimate partner violence is associated with cessation of EBF in the early postnatal period, in line with previous studies [33,57,58]. In most Australian healthcare centres, routine screening of women in the antepartum and postpartum periods for depression [13,55] and IPV [33,59] is conducted in an effort to promptly identify at-risk mothers, ensure referral for clinical assessment, follow-up, and support. Our findings suggest that sustained advocacy is required for perinatal depression and IPV screening among Australian CALD women to improve EBF.

Past studies conducted in Australia and internationally, have shown that cigarette smoking is associated with suboptimal breastfeeding outcomes [18,60]. Similarly, the present study shows that cigarette smoking is associated with cessation of EBF in the early postnatal period among CALD mothers. Evidence suggests that while there is an association between cigarette smoking and suboptimal breastfeeding, the evidence for a plausible biological mechanism is weak [61]. Nevertheless, a recent review indicated that smoking reduces the protective effects of breast milk and also negatively modifies the infant’s response to breastfeeding and breast milk [62]. A possible explanation for this association among smokers may stem from a lack of motivation to breastfeed (i.e., less likely to have breastfeeding intention or to initiate breastfeeding) and/or less desire to seek help with breastfeeding challenges compared to non-smokers [60,61,63]. Australia has legislated a range of anti-smoking strategies (including restrictions on advertising, incremental taxation and regulation) to reduce smoking in the community. These initiatives have been shown to be effective in reducing experimentation and uptake of smoking among young people and overall smoking rates across all socio-demographics in Australia [63]. Sustained implementation of these initiatives will have positive impacts on breastfeeding outcomes among CALD Australian mothers in the short- and long-term.

There is robust evidence in the literature which indicates that health facility birthing, mode of birthing, and professional assistance received during birthing are critical to a mother’s breastfeeding initiation and continuation [18,64]. This possibly reflects the key role the Baby Friendly Hospital Initiative (BFHI) plays in promoting, protecting, and supporting optimal breastfeeding in health facilities. Our study found that caesarean birthing was associated with cessation of EBF in the early postpartum period, which is similar to findings from previous studies [18,19]. Among Australian states and territories, evidence indicates that NSW has the lowest number of BFHI certified maternity centres, with implications for maternal breastfeeding experiences [18]. It is anticipated that the Australian National Breastfeeding Strategy 2019 and beyond will provide a roadmap for improving breastfeeding outcomes among CALD mothers in Australia, including increasing BFHI certified maternity centres in NSW [65]. The study showed that the odds of stopping EBF in the immediate postpartum period were lower among mothers from all major nationality groups compared to their counterparts in the reference group (Oceania). Future studies that focus on breastfeeding outcomes among sub-groups within the CALD populations may be warranted, as they would provide more insights for cultural care.

Limitations of this study are discussed. First, the cessation of breastfeeding in the early postnatal period was based on self-report, potentially leading a recall and/or measurement bias. The implication of this is that it may have resulted in an underestimation or overestimation of the relationship between the risk factors and the outcome. Second, the study was unable to assess or adjust for all potential determining factors (e.g., prematurity, level of support received postnatally, multi-parity, or partner education) as these may also affect the observed results. Third, we were unable to distinguish mothers who ceased EBF in the first, second, third, or fourth week postpartum. This analysis would have provided detailed information on early cessation of EBF postnatally among CALD Australian mothers. Fourth, the small sample size of mothers who indicated no prenatal breastfeeding intention or who were undecided may account for the large effect size. Fifth, the non-use of maternal and child health data from private healthcare and other local health districts in Sydney is a limitation in the present study. Finally, the unavailability of longitudinal data on EBF from 1 to 6 months postnatally and the use of secondary data were additional limitations in this study. Despite these limitations, the study provides breastfeeding data from CALD Australian women to inform population-level interventional strategies.

## 5. Conclusions

Our study suggests that many CALD mothers have definite intention to breastfeed, and most mothers practise skin-to-skin contact, EBF at delivery and at discharge from hospital post-birth. Notably, EBF remained high in the early postnatal period (within 4-weeks postpartum) among CALD Australian mothers. A lack of maternal prenatal breastfeeding intention and partner support, antenatal and postnatal depressive symptoms, psychosocial intimate partner violence, caesarean birthing, low socioeconomic status, and young maternal age (<20 years) were associated with the cessation of EBF in the early postnatal period among CALD Australian mothers. Our study provides insight into breastfeeding practices among CALD Australian mothers to inform targeted initiatives, especially those identified to be at risk of early cessation of EBF.

## Figures and Tables

**Table 1 nutrients-11-01611-t001:** Characteristics of the study population (*N* = 25,407).

Variables	N	(%)
**Sociodemographic factors**		
**Maternal age group**	25,407	
20–34 years	23,812	93.7
<20 years	124	0.5
≥35 years	1471	5.8
**SES category**	23,786	
Low	12,548	52.8
Middle	9377	39.4
High	1861	7.8
**Smoking status**	23,750	
No	23,170	97.6
Yes	580	2.4
**Supportive partner**	21,672	
Yes	21,061	97.2
No	611	2.8
**Major nationality group**	25,407	
Oceania	1481	5.8
North-West Europe	384	1.5
Southern-Eastern Europe	1316	5.2
North Africa and The Middle East	5846	23.0
South-East Asia	6222	24.5
North-East Asia	3512	13.8
Southern and Central Asia	5127	20.2
Americas	634	2.5
Sub-Saharan Africa	885	3.5
**Health factors**		
**Antenatal health problems**	24,603	
No	20,517	83.4
Yes	4086	16.6
**Psychosocial intimate partner violence**	21,583	
No	21,238	98.4
Yes	345	1.6
**Physical intimate partner violence**	21,523	
No	21,240	98.7
Yes	283	1.3
**Type of delivery**	25,378	
Normal vaginal	15,004	59.1
Assisted vaginal	2924	11.5
Caesarean section	7450	29.4
**Antenatal depressive symptoms**	20,560	
EPDS ≤ 9	16,972	82.6
EPDS 10–12	2078	10.1
EPDS ≥ 13	1510	7.3
**Postnatal depressive symptoms**	19,342	
EPDS ≤ 9	17,425	90.1
EPDS 10–12	1194	6.2
EPDS ≥ 13	723	3.7

N = Sample size; EPDS: Edinburgh Postnatal Depression Scale; SES: Socioeconomic status.

**Table 2 nutrients-11-01611-t002:** Prevalence of breastfeeding among culturally and linguistically diverse (CALD) mothers from South Western Sydney and Sydney Local Health Districts in Sydney, Australia, 2014–2016 (*N* = 25,407).

Variables	Breastfeeding Intention	Skin-to-Skin Contact	EBF at Delivery	EBF at Discharge	EBF at 1–4 Weeks
	n (%)	n (%)	n (%)	n (%)	n (%)
**Study outcomes**					
Yes	21,232 (94.7)	16,022 (81.2)	19,355 (90.7)	22,555 (93.3)	19,244 (91.4)
No	629 (2.8)	3697 (18.8)	1973 (9.3)	1629 (6.7)	1809 (8.6)
Undecided	572 (2.5)	-	-	-	-
**Socio-demographic factors**					
**Maternal age group**					
20–34 years	21,048 (93.8)	18,651 (94.6)	20,023 (93.9)	22,690 (93.8)	1628 (90.0)
<20 years	103 (0.5)	113 (0.6)	112 (0.5)	1375 (5.7)	14 (9.2)
≥35 years	1282 (5.7)	955 (4.8)	1193 (5.6)	119 (0.5)	167 (0.8)
**SES category**					
High	10,304 (63.4)	9778 (52.9)	10,411 (52.1)	11,877 (52.5)	1109 (63.1)
Middle	7962 (32.8)	7268 (39.3)	7958 (39.8)	8922 (39.5)	588 (33.4)
Low	1598 (3.8)	1427 (7.7)	1612 (8.1)	1817 (8.0)	62 (3.5)
**Smoking status**					
No	21,860 (97.6)	17,937 (97.3)	19,520 (97.7)	22,207 (97.6)	1532 (92.9)
Yes	543 (2.4)	437 (2.4)	467 (2.3)	540 (2.4)	117 (7.1)
**Supportive partner**					
Yes	20,501 (97.2)	16,457 (97.2)	17,773 (97.2)	20,190 (97.3)	1454 (95.7)
No	585 (2.8)	481 (2.8)	508 (2.8)	569 (2.7)	66 (4.3)
**Major nationality group**					
Oceania	1237 (5.5)	1193 (6.1)	1247 (5.9)	1368 (5.7)	221 (12.2)
North-West Europe	360 (1.6)	326 (1.7)	343 (1.6)	373 (1.5)	18 (1.0)
Southern-Eastern Europe	1177 (5.3)	1028 (5.2)	1146 (5.4)	1262 (5.2)	128 (7.08)
North Africa and The Middle East	5100 (22.7)	4649 (23.6)	4901 (23.0)	5526 (22.8)	612 (33.8)
South-East Asia	5437 (24.2)	4958 (25.1)	5235 (24.6)	5868 (24.3)	487 (26.9)
North-East Asia	3150 (14.0)	2837 (14.4)	3048 (14.3)	3431 (14.2)	150 (8.3)
Southern and Central Asia	4636 (20.7)	3638 (18.5)	4158 (19.5)	4914 (20.3)	116 (6.4)
Americas	552 (2.5)	439 (2.2)	526 (2.5)	607 (2.5)	42 (2.3)
Sub-Saharan Africa	784 (3.5)	651 (3.3)	724 (3.4)	835 (3.5)	35 (1.9)
**Health factors**					
**Antenatal health problems**					
No	17,847 (82.4)	15,821 (83.0)	17,334 (83.7)	19,514 (83.3)	1496 (85.1)
Yes	3819 (17.6)	3245 (17.0)	3370 (16.3)	3916 (16.7)	263 (14.9)
**Psychosocial intimate partner violence**					
No	20,657 (98.4)	19,618 (98.4)	17,919 (98.4)	20,360 (98.5)	1517 (97.9)
Yes	335 (1.6)	273 (1.6)	290 (1.6)	316 (1.5)	33 (2.1)
**Physical intimate partner violence**					
No	20,663 (98.7)	16,616 (98.7)	17,911 (98.7)	20,356 (98.7)	1522 (98.1)
Yes	270 (1.3)	213 (1.3)	243 (1.3)	265 (1.3)	30 (1.9)
**Type of delivery**					
Normal vaginal	13,356 (59.6)	14,903 (75.7)	14,126 (66.3)	14,432 (59.7)	988 (54.7)
Assisted vaginal	2581 (11.5)	2878 (14.6)	2452 (11.5)	2759 (11.4)	163 (9.0)
Caesarean section	6473 (28.9)	1913 (9.7)	4730 (22.2)	6968 (28.8)	655 (36.3)
**Antenatal depressive symptoms**					
EPDS ≤ 9	16,543 (72.6)	13,406 (83.3)	14,392 (82.9)	16,259 (82.6)	1164 (79.6)
EPDS 10–12	2013 (10.1)	1577 (9.8)	1751 (10.1)	1992 (10.1)	146 (10.0)
EPDS ≥ 13	1465 (7.3)	1121 (6.9)	1222 (7.0)	1433 (7.3)	152 (10.4)
**Postnatal depressive symptoms**					
EPDS ≤ 9	15,451 (90.2)	13,538 (90.6)	14,739 (90.5)	16,628 (90.2)	1,489 (88.9)
EPDS 10–12	1055 (6.2)	880 (5.9)	973 (6.0)	1139 (6.2)	94 (5.6)
EPDS ≥ 13	633 (3.7)	523 (3.5)	579 (3.5)	677 (3.6)	92 (5.5)

n: cases; antenatal health problems included diabetes mellitus and/or hypertension; EBF at delivery was defined as infants who received only breast milk within the first 24 h post-delivery; EBF at discharge was measured as infants who received only breast milk in the 24 h preceding discharge from the maternity unit; EPDS: Edinburgh Postnatal Depression Scale; SES: Socioeconomic status.

**Table 3 nutrients-11-01611-t003:** Determinants of exclusive breastfeeding cessation in the early postnatal period among CALD mothers in South Western Sydney and Sydney Local Health Districts, 2014–2016 (*N* = 25,407).

Study Factors	Complete Case Analysis				Multiple Imputation Analysis *			
	Unadjusted OR (95% CI)	*p* Value	Adjusted OR (95% CI) (a)	*p* Value	Unadjusted OR (95% CI)	*p* Value	Adjusted OR (95% CI) (a)	*p* Value
**Antenatal breastfeeding intention**								
Yes	1.00		1.00		1.00		1.00	
No	31.11 (25.41–38.10)	<0.001	23.76 (18.63–30.30)	<0.001	31.11 (29.90–32.37)	<0.001	27.13 (26.04–28.27)	<0.001
Undecided	7.01 (5.69–8.64)	<0.001	6.15 (4.74–7.98)	<0.001	7.00 (6.72–7.30)	<0.001	6.38 (6.11–6.66)	<0.001
Sociodemographic factors								
**Supportive partner**								
Yes	1.00		1.00		1.00		1.00	
No	1.81 (1.38–2.36)	<0.001	1.69 (1.20–2.38)	0.003	1.66 (1.58–1.74)	<0.001	1.60 (1.52–1.69)	<0.001
**Socio-economic status**								
Low	1.00		1.00					
Middle	0.67 (0.61–0.75)	<0.001	0.85 (0.72–0.99)	0.044	0.67 (0.65–0.68)	<0.001	0.84 (0.82–0.86)	<0.001
High	0.36 (0.28–0.47)	<0.001	0.48 (0.32–0.71)	<0.001	0.36 (0.34–0.38)	<0.001	0.59 (0.56–0.63)	<0.001
**Maternal age group**								
20–34 years	1.00		1.00		1.00		1.00	
<20 years	1.76 (1.48–2.08)	<0.001	1.72 (1.37–2.15)	<0.001	1.75 (1.69–1.81)	<0.001	1.75 (1.69–1.81)	<0.001
≥35 years	1.79 (1.02–3.15)	0.044	1.91 (0.93–3.93)	0.078	1.78 (1.60–1.99)	<0.001	1.61 (1.44–1.80)	<0.001
**Smoking status**								
No	1.00		1.00		1.00		1.00	
Yes	4.29 (3.45–5.33)	<0.001	3.39 (2.56–4.49)	<0.001	4.28 (4.10–4.47)	<0.001	3.81 (3.65–3.99)	<0.001
**Major nationality group**								
Oceania	1.00				1.00			
North-West Europe	0.25 (0.15–0.41)	<0.001	0.49 (0.25–0.97)	0.043	0.25 (0.23–0.28)	<0.001	0.47 (0.43–0.52)	<0.001
Southern-Eastern Europe	0.57 (0.45–0.73)	<0.001	0.73 (0.53–1.00)	0.051	0.57 (0.55–0.60)	<0.001	0.71 (0.68–0.74)	<0.001
North Africa and The Middle East	0.67 (0.56–0.79)	<0.001	0.65 (0.51–0.82)	<0.001	0.67 (0.65–0.69)	<0.001	0.64 (0.62–0.67)	<0.001
South-East Asia	0.45 (0.38–0.53)	<0.001	0.46 (0.36–0.58)	<0.001	0.45 (0.43–0.46)	<0.001	0.47 (0.45–0.49)	<0.001
North-East Asia	0.24 (0.19–0.31)	<0.001	0.41 (0.29–0.56)	<0.001	0.24 (0.23–0.26)	<0.001	0.43 (0.41–0.45)	<0.001
Southern and Central Asia	0.11 (0.09–0.14)	<0.001	0.18 (0.13–0.24)	<0.001	0.11 (0.11–0.12)	<0.001	0.14 (0.14–0.15)	<0.001
Americas	0.37 (0.26–0.53)	<0.001	0.40 (0.24–0.66)	<0.001	0.37 (0.35–0.40)	<0.001	0.41 (0.38–0.44)	<0.001
Sub-Saharan Africa	0.21 (0.14–0.31)	<0.001	0.25 (0.15–0.41)	<0.001	0.21 (0.20–0.23)	<0.001	0.24 (0.23–0.26)	<0.001
Health factors								
**Antenatal depressive symptoms**								
EPDS ≤ 9	1.00		1.00		1.00		1.00	
EPDS 10–12	1.03 (0.86–1.23)	0.783	1.10 (0.88–1.36)	0.398	1.02 (0.99–1.06)	0.161	1.00 (0.97–1.04)	<0.001
EPDS ≥ 13	1.59 (1.33–1.91)	<0.001	1.50 (1.20–1.89)	<0.001	1.59 (1.53–1.64)	<0.001	1.57 (1.51–1.62)	<0.001
**Postnatal depressive symptoms**								
EPDS ≤ 9	1.00		1.00		1.00		1.00	
EPDS 10–12	0.911 (0.73–1.13)	0.402	0.97 (0.73–1.30)	0.860	0.92 (0.88–0.96)	<0.001	1.00 (0.96–1.04)	0.885
EPDS ≥ 13	1.61 (1.28–2.01)	<0.001	2.07 (1.55–2.77)	<0.001	1.57 (1.50–1.63)	<0.001	1.66 (1.59–1.73)	<0.001
**Psychosocial intimate partner violence**								
No	1.00		1.00		1.00		1.00	
Yes	1.47 (1.02–2.12)	0.041	1.66 (1.10–2.53)	0.017	1.46 (1.36–1.57)	<0.001	1.50 (1.39–1.61)	<0.001
**Physical intimate partner violence**								
No	1.00		1.00		1.00		1.00	
Yes	1.57 (1.06–2.31)	0.023	1.49 (0.94–2.38)	0.093	1.56 (1.45–1.69)	<0.001	1.62 (1.50–1.75)	<0.001
**Type of delivery**								
Normal vaginal	1.00		1.00		1.00		1.00	
Assisted vaginal	0.79 (0.67–0.94)	0.008	0.90 (0.72–1.14)	0.386	0.79 (0.76–0.82)	<0.001	0.96 (0.88–0.92)	0.078
Caesarean section	1.36 (1.23–1.51)	<0.001	1.35 (1.18–1.55)	<0.001	1.36 (1.33–1.39)	<0.001	1.48 (1.45–1.52)	<0.001
**Pre-existing maternal health problems**								
No	1.00		1.00		1.00		1.00	
Yes	0.87 (0.76–1.00)	0.054	1.02 (0.83–1.07)	0.790	0.87 (0.85–0.89)	<0.001	0.95 (0.93–0.98)	0.004

(a): adjusted for maternal body mass index, gender of the baby, maternal alcohol intake, Aboriginality and birthing facility location, as well as socioeconomic and health factors. * Sensitivity analyses following multiple imputations for missing values; pre-existing maternal health problems included diabetes mellitus and/or hypertension.

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
