# Peer review of "Determinants of Exclusive Breastfeeding Cessation in the Early Postnatal Period among Culturally and Linguistically Diverse (CALD) Australian Mothers"

_nutrients, 2019, doi:10.3390/nu11071611_

Author Response

Reviewer’s feedback:

Thank you for giving me this opportunity to read your paper. I have listed my points on the following pages that I think need to be addressed prior to publication.

It can be difficulty to spot spelling and grammatical errors when you spend so long writing a paper. I would recommend asking a friend, who is not invested in the study, to check the paper for spelling and grammatical errors prior to re-submission. I have given two examples from the ‘Introduction’ that requires editing.

Response:

The revised manuscript has been double-checked for spelling and grammatical errors by co-authors who native English speakers.

Also, check your references. For example, Reference 9 states the cost benefit of breastfeeding to the state and not the household and the link for reference 10 doesn’t work.

Response

References now edited.

Any text written in red and highlighted in yellow are my edits to the paper. These are just suggestions.

Response

Points appreciated and now reflected in the revised manuscript.

Abstract

Page 2, Line 52: From reading your paper I would interrupt the findings to show that women of CALD backgrounds in Australia do not need breastfeeding interventions that are tailored to their specific needs as they have higher EBF rates to non-CALD mothers!

Response

We agree. The text has been edited for clarity.

Introduction

Page 2, Line 56-59: I would include the time period in this sentence– exclusively breastfeeding is recommend for 6 months. You do state this on line 62 but this information can be incorporated into the opening sentence.

For example:

“Global health organisations (such as the World Health Organization and the United Nations Children’s Fund, WHO/UNICEF) recommend exclusive breastfeeding (EBF), for the first six months of life. EBF is defined as providing the infant with human breastmilk only, and when needed oral rehydration solution, or drops/syrups of vitamins, minerals or medicines  infant's consumption of human milk with no supplementation with water, nonhuman milk, juice and/or foods, except for vitamins, minerals and medications..”

Response

Thank you. The text has been edited for clarity (Page 2, Paragraph 1).

Page 2, Line 60-62: I would edit this sentence, “The breast milk has a high proportion..” I would remove ‘The’ and if you use the word ‘high’ you need to give a comparative (high to what?)

What about: “Breastmilk is a species-specific nutrient aimed to promote the desired growth and development of infants.”

Response

Thank you. The text has been edited for clarity (Page 2, Paragraph 1).

Page 2, Line 79: You state “…determinants of suboptimal EBF among CALD subgroups…” but need to define what you consider suboptimal to be. To my knowledge no country meets the WHO recommendation of 90% exclusively breastfed infants.

Response

The text has been edited for clarity (Page 2, Paragraph 1).

Methods

Page 3, Line 96: I would state if the data used was information that is routinely collected as part of standard care provided during pregnancy and postnatal period.

Response

Thank you. The text has been incorporated into the revised manuscript as suggested by the reviewer (Page 3, Paragraph 1).

Is the data collected for all mothers regardless if they had private or public healthcare? If the data is only of mothers that use public healthcare than this a limitation of your study and needs to be expanded on in the discussion.

Response

As noted in the original manuscript we only used data from public facilities. Additional text has been incorporated into the Discussion section of the revised manuscript (Line 317).

Page 3, Line 106: You state that the potential study sample includes the ‘most socioeconomically disadvantaged populations’ but does it include CALD Australian mothers?

Response

The text on socio-economic disadvantaged is based on data from the Australian Bureau of Statistics. Yes, it does include CALD and non-CALD populations in Sydney. A reference has been incorporated for clarity.

Also, how representative is your study sample, from Sydney (which to my knowledge is one of the most expensive cities in the world), of all CALD Australian mothers? Is this a limitation of your study?

Response

Yes, it may be a limitation in the study – this has been included in the Discussion section of the revised manuscript (Line 317).

Page 3, Line 118: Need to state when (at point in the pregnancy) the mother was asked her breastfeeding intention.

Response

This information was noted in the original manuscript (Page 3, Paragraph 1):

Antenatal information that included socio-demographic characteristics, history of any previous pregnancy and mother’s breastfeeding intention were collected by qualified midwives at the first prenatal care visit.

Also, when they were asked about breastfeeding intention was that their intention to exclusive or/ and any breastfed?

Response

Mothers were asked about the broad concept of “breastfeeding intention”, not specific to exclusive and/or any other breastfeeding option. This is consistent with previous studies as noted in the original manuscript (Page 2, Paragraph 3 and Discussion section, Paragraph 3).

Page 3, Line 119: Define ‘risk factors’.

Response

The text has been clarified in the revised manuscript (Page 3, Paragraph 2).

Page 3, Line 121-122: You state “..while EBF at discharge was measured as the proportion of infants who received only breast milk 24 hours preceding discharge from the maternity unit.”

Therefore, potentially, you have included mixed feeders in your sample and this can impact on your results. Infant formula top-ups are a known risk factor for shorter breastfeeding duration. There are also differences between mothers/ infants that do and do not use infant formula in the maternity hospital.

Response

We did not anticipate that mixed feeding information would have been included in the study sample as these data were collected and entered into the MCH database be qualified midwives who have a good understanding of exclusive breastfeeding and other infant feeding options. In the database, variables relating to mixed feeding and other infant feeding options were also collected and noted but were not included in the present study. The 24-hour recall approach is consistent with the World Health Organisation definitions for assessing infant and young child feeding. We agree with the reviewer that there is a possibility for recall bias given this approach, and we noted this limitation in the Discussion section of the original manuscript (Line 307-309).

This point also stands for the postnatal assessments, as mothers were asked about feeding habits in the past 24 hours only.

Response

As noted above, no action was taken in the revised manuscript.

Page 3, Line : Did you collect information on infants being admitted to neonatal units during maternity stay or gestational age at delivery? Both factors are known to influence EBF. If using routine collected data these variables should be included. Are CALD Australian mothers more likely to give birth prematurely and have their babies admitted to a neonatal unit?

Response

Information on admission into neonatal units or gestational age at delivery was not available to the investigators. We acknowledged in the limitation section of the original manuscript our inability to assess for all potential factors associated with EBF – this text has now been clarified in the revised manuscript (Line 321).

Therefore, is it that CALD mothers need different interventions or is it they are more likely to experience the risk factors for early BEF cessation?

Response

As noted above, we were not privy to the information on birthing interventions and CALD mothers at the time. Nevertheless, we also want to note that one of our colleagues – an obstetric registrar – is currently investigating the impact of birthing interventions in both CALD and non-CALD women.

From the descriptive you have provided of the dataset I feel that you should have more variables to explore – why have you just focused on these.

Response

We agree that more data may have been explored; however, our study factors were based on current data availability.

Page 3, Line 135: Need to justify why you categorised continuous variables. Given, as you said, that there is limited data on breastfeeding among CALD mothers how do you know you picked the right cut-off points that are specific to CALD mothers?

Response

Cut-offs were selected based on current policy interventions and past studies. For example, the EPDS cut-off points used were based on current New South Wales Health recommendations. Additional information on the EPDS in this context is published elsewhere (Ogbo et al., 2018).

Reference

Ogbo et al. Determinants of antenatal depression and postnatal depression in Australia. BMC Psychiatry. 2018;18:49.

Page 3, Line 142: When was the “Edinburgh Postnatal Depression Scale” done? Were mothers referred to other services based on their score?

Response

The text has been clarified in the revised manuscript (Page 3, Paragraph 2). Antenatal information that included socio-demographic characteristics, history of any previous pregnancy, probable depression based on the Edinburgh Postnatal Depression Scale (EPDS) and mother’s breastfeeding intention were collected by qualified midwives at the first prenatal care visit.

Yes, mothers were referred to other services based on their score. Information relating to the use of the EDPS in this context is published elsewhere (Ogbo et al, 2018 & Eastwood et al, 2017).

·         Ogbo FA, Eastwood J, Hendry A, Jalaludin B, Agho KE, Barnett B, et al. Determinants of antenatal depression and postnatal depression in Australia. BMC Psychiatry. 2018;18(1):49.

·         Eastwood J, Ogbo FA, Hendry A, Noble J, Page A, Early Years Research Group. The impact of antenatal depression on perinatal outcomes in Australian women. Plos One. 2017;12(1):e0169907

A mother with a high score who is referred will differ to a mother with a score who is not referred.

Response

We agreed with the reviewer that there may be differences in mothers who are referred for further assessment and those who are not. However, we believe that this is beyond the scope of the present study. We note that a separate manuscript that considers antenatal and postnatal depression using the EPDS has been published elsewhere (Ogbo et al., 2018). Also, another manuscript which examined who should be referred and who shouldn’t be referred based on the EPDS cut-offs have been accepted for publication in the BMC Pregnancy and Childbirth.

Reference

Ogbo et al. Determinants of antenatal depression and postnatal depression in Australia. BMC Psychiatry. 2018;18:49.

Page 3, Line 132: What is your inclusion/ exclusion criteria? I know you are looking at CALD mothers but how did you identify them from the dataset?

As per Department of Health statements: “Culturally and linguistically diverse (CaLD): Groups and individuals who differ according to religion, race, language or ethnicity, except those whose ancestry is Anglo Saxon, Anglo Celtic, Aboriginal or Torres Strait Islander. CaLD-related data or CaLD data: data variables or parameters that measure those attributes of persons that relate to their cultural or language background. Language is only mentioned once in your paper and this is in the introduction.

Response

During the first antenatal visit, women were asked to identify whether they belong to CALD, non-CALD or Aboriginal or Torres Strait Islander subpopulation based on the definition, and this information was entered into the database. This information has now been noted in the revised manuscript (Page 3, Paragraph 1).

Results

Page 5, Table 1: Ensure the correct symbols are used. For example, you have “>35 years” when it should be “≥35 years”

Response

Thank you for the observation. Text now edited.

Table 1: Rates of EBF increase between delivery and discharge. Also, rates of EBF remain higher at 1-4 weeks postnatal compared to at delivery. I would suggest you clean the data – as rates of exclusive breastfeeding can only ever go down and not up as the infant ages.

Response

Thank you for bringing this to our attention. If there was an increase in EBF prevalence between delivery and discharge, it may be due to a number of reasons, including i) macrosomia and its subsequent management; ii) potential effects of interventional delivery; or iii) a lack of immediate milk let down. Nevertheless, additional analysis of the data (ie, the estimation of the confidence interval and P-value around the estimates, data not shown) suggested that there was no significant difference between EBF prevalence at delivery and discharge. This information has been noted in the revised manuscript (Results, below Table 1, Paragraph 2).

Table 1: Can you split Table 1 into two tables?

Table 1 – just the outcome data

Table 2 – univariate analysis results of the exposure and outcome data. This way the reader can see not just the number and percentage but also the p-value. The reader can know see why you picked the variables that you did for the multivariate analysis. 

Response

Point appreciated and now reflected in the revised manuscript (Table 1 & 2).

Page 7, Lines 190-193 (and Table 1): The 2010 Australian national infant feeding survey reported that “A total of 90.4% of infants initiated exclusive breastfeeding (that is, their first feed was breastmilk or equivalent) (see Table 3.7). Among all infants, 61.4% were exclusively breastfed for less than 1 month”. You are reporting that 93.3% of CALD infants left the maternity hospital exclusively breastfeeding and by 1-4 weeks 91.4% of CALD mothers continue to EBM. Therefore, by your results CALD mothers have much higher breastfeeding rates no non-CALD mothers! This is suggesting measurement (i.e. dataset has not accurately captured the required information, error in coding and/or statistical analysis) and selection bias (your dataset is not representative of the study population). I would re-peat the analysis and include non-CALD mothers as your comparison group. 

Response

As noted by the reviewer, the Australian National Infant Feeding Survey was conducted almost a decade ago, and during this time, there have been a number of targeted interventions to improve breastfeeding outcomes among women at the district, state and national levels in Australia.

In this study, the authors have not suggested that these results are representative of the general Australian population. But that these results are relevant to CALD women living in Sydney, New South Wales.

Furthermore, selection bias is not an issue in the study as we used available and routinely collected MCH data for two specific Local Health Districts – we did not perform sampling to obtain the sample size. As noted above, breastfeeding information were collected by qualified midwives in those health districts.

Finally, we believe that national data may not be comparable to discrete efforts as these entities are not comparable. As noted in the manuscript, specific quantitative studies that focused on CALD populations are limited, hence, this study aims to highlight key information relating to breastfeeding among CALD Australian women living in Sydney.

Page 194-199: Need to state which result you are presenting – adjusted complete case analysis.

Response

Point appreciated and now reflected in the revised manuscript (Results section).

Also, this section is very difficult to read and I would avoid long sentences were possible. I would re-edit these sentences.

For example: “In The complete case multivariate analysis multivariate The study showed that mothers who indicated no prenatal breastfeeding intention or who were undecided about breastfeeding during pregnancy were more likely to cease EBF in the early postnatal period compared to those who indicated prenatal breastfeeding intention (Odds Ratio [OR]=23.76, 95%CI 18.63-30.30, P<0.001) when controlling for X, X and X. Mothers who were undecided in their breastfeeding intention had over six times the odds of ceasing EBF compared to mothers who intended to EBF (OR=6.15, 95%CI 4.74-7.98, P<0.001, for those undecided) when controlling for X, X and X. [Table 2].”

Response

Point appreciated and now reflected in the revised manuscript (Results section). We noted that the phrase ‘multivariate’ already indicates adjustment for cofactors, and therefore, was not included in the sentence.

Page 8, Table 2: Need to provide the sample size for each result. How many participants were missing data and therefore excluded from the complete case analysis?

Can just add a row to the table giving sample sizes.

Response

This information (sample size for each variable) has now been incorporated into table 1, also consistent with Reviewer 2 comments below.

Page 9, 204-206: You state “Mothers who reported not having a supportive partner during pregnancy were more likely to stop EBF in the early post-birth period compared to those who reported receiving support from their partner during pregnancy (OR=1.69, 95%CI: 1.20-2.38, P=0.003).

Yet, in table 2 for complete case univariate analysis the OR is 4.29 and for adjusted complete case analysis the OR is 3.39; for univariate multiple imputation analysis the OR is 1.66 and for adjusted multiple imputation analysis the OR is 1.60.

Where has the 1.69 come from? Again, this highlights the importance of stating which result you are referring to.

Response

Thank you for the observation. It was an error on our part, where smoking data were mistakenly entered into the partner support row. The text has been edited in the revised manuscript (Table 3)

Discussion

Page 10, Line 227-228: “Notably, EBF remained high in the early postnatal period among CALD Australian mothers (91.4%), higher than the EBF prevalence for the general population in this context (62%) [18]”. – This is a big finding, need to discuss and explore possible reasons for this result

Response

We have deleted this point given that a manuscript which compared breastfeeding patterns among CALD and non-CALD women based on a 5-year MCH data is been developed by a Public Health Trainee.

Page 11, Line 302: Another key limitation is that this is a secondary data analysis.

Response

Point appreciated and now reflected in the revised manuscript (Limitation section).

Conclusion

Need to give specific, and workable, recommendations to practice, research or policy on your findings. Your comments are too vague, for example “Our finding highlights the importance of routine perinatal depression screening in Australia” – is screening only done in Sydney? And, if you screen you need to action. So, from your results at point should action occur (i.e what is the cut-off score) and what action should take place

Response

We feel the text was expanded further: Our finding highlights the importance of routine perinatal depression screening in Australia [54]. Furthermore, the present study also showed that self-reported psychological intimate partner violence was associated with cessation of EBF in the early postnatal period, in line with previous studies [55-57]. In most Australian healthcare centres, routine screening of women in the antepartum and postpartum periods for depression [13, 54] and IPV [58, 59] is conducted in an effort to promptly identify at-risk mothers, ensure referral for clinical assessment, follow-up and support. Our findings suggest that sustained advocacy is required for perinatal depression and IPV screening among Australian CALD women to improve EBF.

Publications on the impact and determinants of perinatal depression screening, including information on specific, workable, and policy implications are published elsewhere (Eastwood et al 2017 and Ogbo et al, 2018). The text has now been clarified accordingly in the revised manuscript (Discussion section, Paragraph 4).

·         Ogbo FA, Eastwood J, Hendry A, Jalaludin B, Agho KE, Barnett B, et al. Determinants of antenatal depression and postnatal depression in Australia. BMC Psychiatry. 2018;18(1):49.

·         Eastwood J, Ogbo FA, Hendry A, Noble J, Page A, Early Years Research Group. The impact of antenatal depression on perinatal outcomes in Australian women. Plos One. 2017;12(1):e0169907

Reviewer 2 Report

Please see file

Author Response

Reviewer 2:

Determinants of exclusive breastfeeding cessation in the early postnatal period among culturally and linguistically diverse (CALD) Australian mothers

Overall comments:

This study investigated the determinants of exclusive breastfeeding (EBF) cessation in the early postnatal period (1-4 weeks post birth) among culturally and linguistically diverse (CALD) Australian mothers in Sydney, Australia by using linked maternal and child health data from two Local Health Districts in Australia (N=25,407). The study found a high EBF rate among CALD Australian mothers in the early postnatal period (91.4%). A lack of maternal prenatal breastfeeding intention and partner support, antenatal and postnatal depressive symptoms, psychosocial intimate partner violence, caesarean delivery mode, low socioeconomic status and young maternal age (<20 years) were found to be significantly associated with the cessation of EBF in the early postnatal period among CALD Australian mothers in this study.

This manuscript is a good contribution in making use of health-linked data and its findings are important to those who are with closely related research interests. Generally, the manuscript is well written, and its description reasonably succinct and straightforward to read. Here are some detailed comments for the authors to consider.

Response

Thank you for the comment.

Specific comments:

Line 94-98: 2. Materials and Methods

It is not clear how the study sample (CALD mothers) was chosen from the maternal and child health data of all live births in public health facilities in SLHD and SWSLHD between 2014 and 2016? The authors need to provide clear definition of CALD population and sampling information here.

Response

Point appreciated and now reflected in the revised manuscript (Page 3, Paragraph 1). Sampling was not required as we used all available MCH data.

Line 112: 2.3.Outcome variables

mother’s breastfeeding intention, skin-to-skin contact”: it is known from literature that these two variables have some impacts on EBF and have been studies as a predictor of EBF in a number of papers. The authors should make a justification why they were used as outcome variables in this study. In addition, the authors need to explain the reason for changing the role of “mother’s breastfeeding intention” from a dependent variable to an independent (a potential factor) variable later in Section 3.2 (Line 194).

It is not clear whether the role of “skin-to-skin contact” was changed and whether it was included in the multiple regression analysis.

Response

Additional information has been incorporated into the revised manuscript (Page 4, Paragraph 2). The rationale for selecting the study variables was noted in the original manuscript (Study factors section): The exposure variables were broadly categorised into socio-demographic and health factors, which were selected based on previous studies [18, 19, 22, 23] and data availability.

Information as to why breastfeeding intention was used as a potential factor has been clarified in the revised manuscript (Page 4, Paragraph 2): Maternal breastfeeding intention was also considered a potential factor for the cessation of EBF in the early post-birth period based on past studies.

The study outcomes (including skin-to-skin contact) were not included in the multivariate analyses. As noted in original manuscript, cessation of EBF was the main outcome variable included in the multiple regression analysis.

Furthermore, the authors need to explain how to measure/obtain these two variables from the linked database (same as what they have explained clearly for EBF at delivery, EBF at discharge, EBF in the early postnatal visit).

Response

Point appreciated and now reflected in the revised manuscript (Page 3, Paragraph 5).

Line 153: 2.5. Statistical analysis

“Multivariate logistic regression analyses” “Multivariate imputation”: “Multivariate” should be replaced by “Multiple” (or Multivariable”). Statistically “Multivariate” refers to multivariate analysis, which is not the case for the present study, and suggest correcting.

Response

Thank you for the observation. This was an error on our part and has now been corrected in the revised manuscript.

As “This study aimed to investigate the determinants of EBF cessation in the early postnatal period among CALD Australian mothers”, the authors should provide the details of the statistical modelling on identifying the determinants. For example, after “a series of univariate regression models to examine the association between each study factor and cessation of EBF in the early postnatal period”, how did the authors choose the candidate study factors (variables) into the multiple logistic regression analysis? Did authors use any stepwise (backward elimination or forward selection) modelling strategy to achieve the final model presented in Table 2? Did the authors concern about any issue with multicollinearity?

Response

We have re-written the statistical section to make the information clearer to readers. We also note that our Institute Senior Biostatistician (Dr Kingsley Agho and a Guest Editor for Nutrients, Breastfeeding Issue) has also reviewed the statistical section of the manuscript. The exposure variables were selected based on previous studies and data availability as noted in the original manuscript.

We did not perform stepwise modelling but based on a priori knowledge and data availability. Yes, we thought about multicollinearity as with our previously published studies (Agho et al, 2016 & Ogbo et al 2019). Multicollinearity was not evident in the current study.

·         Agho, K. E., Osuagwu, U. L., Ezeh, O. K., Ghimire, P. R., Chitekwe, S., & Ogbo, F. A. (2018). Gender differences in factors associated with prehypertension and hypertension in Nepal: A nationwide survey. PloS one, 13(9), e0203278.

·         Ogbo, F. A., Dhami, M. V., Awosemo, A. O., Olusanya, B. O., Olusanya, J., Osuagwu, U. L., ... & Agho, K. E. (2019). Regional prevalence and determinants of exclusive breastfeeding in India. International breastfeeding journal, 14(1), 20.

Regarding the missing values, how many variables retained for the multiple regression analysis were incomplete, what was the range of missing values? Which Stata syntax the authors used for multiple imputations?

Response

The information on missing values has been clarified in Table 1 as used in the complete case analyses.

The mim command was used for the multiple imputations in Stata and has now been noted in the revised manuscript (Statistical analysis section).

Line 185: Table 1

This table is very busy with unclear information and it needs an improvement. To match the authors’ research aim, probably it is more reasonable to break it down into two smaller tables, one for the outcome variables and one for the characteristics of the study sample against the main EBF outcome variable for “Yes %” or “No %”.

Response

Point appreciated and now reflected in the revised manuscript (Table 1 & 2).

Maternal age groups should be listed from the youngest to oldest and the groups are not inclusive: no age of 50 years old (see Line 135: should be matched).

Response

Analysis relating to the maternal age group was guided by past studies which suggest that young and older maternal age were predictors of inappropriate breastfeeding behaviours. The text in Line 135 has been edited.

The authors mentioned that “N: Sample size; n: cases”: this is not clear to the reviewer. “N” stands for the total number of mothers, or stands for those mothers who answered “Yes” to the corresponding outcome variable? Related to SES, if “21,017” is the total number of mothers, how the %s were calculated for High 10,304 (93.6), Middle 7962 (95.2), and Low 1598 (96.9) groups? (Also “Middle” and “Medium” were used interchangeably: should be consistent see Line136).

Response

Point appreciated and now reflected in the entire revised manuscript. Thank you for the observation!

“Major nationality group” was not mentioned in the Section of Study factors (see Line 132), nor in the later multiple regression analysis (see Table 2).

Response

Point appreciated and now reflected in the entire revised manuscript. Thank you for the observation!

As the authors divided all study factors into “socio-demographic and health factors” (see Line133), suggest organising all variables in this Table 1 (also Table 2) by two subheadings: Socio-demographic Factors and Health factors.

Response

Point appreciated and now reflected in the revised manuscript (Tables 1-3).

Line 181: 3.1. Breastfeeding patterns by study factors

The authors need to summarize more results related to other study factors.

Response

Point appreciated and now reflected in the revised manuscript.

Line 194: 3.2. Determinants of exclusive breastfeeding cessation in the early postnatal period

“Odds Ratio [OR]=23.76,95%CI 18.63-30.30”: this is a very large OR with a wide 95%CI, and it probably indicates rare event (small cases) in the “No” group of Breastfeeding Intention in the Complete case analysis. The authors need to show their knowledge and efforts on dealing with this small cases bias problem using an alternative regression approach.

Response

We agree with the reviewer that small cases in the ‘No’ may be due to the large OR. Notably, we noted this information as a limitation in the original manuscript (Line 317):

Fourth, the small sample size for mothers who indicated no prenatal breastfeeding intention or who were undecided may account for the large effect size.

Line 200: Table 2

Information for Socio-economic status is not matched to the description from Line 206 to 209.

In the footnote, the authors should attach the following information:

• –2 log likelihood (deviance) and degree of freedom for the final model (a)

• All variables included in the initial model

• Which regression method used to obtained this final model (a)

Response

As noted above, the text (SES and statistical section) has been edited. The information in the footnote has been edited in line with the statistical analysis method used.

Suggest the authors to add p value of linear trend for each study factor in both unadjusted and adjusted analyses.

Response

We note that the present study did not examine trends over time. Notably, ‘trend’ when estimated often refers to a period of ten years; therefore, it would be extremely confusing for readers if the authors incorporated some “P for trend” in the current study.

The table did not indicate whether the variable “Major nationality group” were included in the regression analyses, however it was listed as one of Study factors in Table 1.

Response

Thank you for the observation. Text now clarified in the revised manuscript.

Line 220-222

Observed relationships between the study factors and the outcome from multiple imputation analyses were largely similar to those in the complete case analysis (Table 2), suggesting that missing data did not substantially impact the observed associations”. However, the authors should have noticed that the effect of age>=35 years, experienced Physical intimate partner violence, and having pre-existing maternal health problems on ceasing EBF became statistically significant based on the imputation data, whereas they were not in the Complete case analysis. In addition, the 95% CI of the adjusted OR for “No” BF intention based on the imputation data is much narrower than the one obtained based on the original data set. The authors need to make a justification about these differences.

Response

Thank you for the observation and suggestion.

Point appreciated and now reflected in the revised manuscript (Results section, Paragraph 4).

Line 249-264:

It looked that the discussion of the effect of SES on EBF was for general population: suggest the authors relating or framing the effect/problem of SES to CALD mothers with supporting literature review.

Response

Point appreciated and now reflected in the revised manuscript (Discussion section, Paragraph 3)

Line 315: 5. Conclusions

The authors should indicate how their findings would add or change existing public health policies/practices related to CALD mothers in Australia

Response

We note that this information has been incorporated into the body of the entire Discussion section of the original manuscript as we want the conclusion to highlight the key points. For example, Discussion section, Paragraph 6: There is robust evidence in the literature which indicates that health facility birthing, mode of birthing and professional assistance received during birthing are critical to a mother’s breastfeeding initiation and continuation [18, 64]. This possibly reflects the key role the Baby Friendly Hospital Initiative (BFHI) plays in promoting, protecting and supporting optimal breastfeeding in health facilities. Our study found that caesarean birthing was associated with cessation of EBF in the early postpartum period, which is similar to findings from previous studies [18, 19]. Among Australian states and territories, evidence indicates that NSW has the lowest number of BFHI certified maternity centres, with implications for maternal breastfeeding experiences [18]. It is anticipated that the Australian National Breastfeeding Strategy 2019 and beyond will provide a roadmap for improving breastfeeding outcomes among CALD mothers in Australia, including increasing BFHI certified maternity centres in NSW [65].

Reviewer 3 Report

This study investigates the determinants of EBF cessation in the early postnatal period among CALD Australian mothers in Sydney. The manuscript is generally well written. The main concern with the study was the use of CALD as one group. It will be interesting to show sub-group results and provide more insights for cultural care.

1.      National data of proportion of CLAD among new mothers and that by regions/ethnicities could be presented in the Introduction. The proportion in the study could be compared to national data in order to shed light on representativeness of the sample.

2.      Rationale of selection of the potential predictors should be described. For example, do you speculate that CLAD mothers have lower social support or higher domestic violence? Do you include maternal work status as one of the predictors?

3.      CLAD Australian mothers are a very diverse group. Do women from different regions differ in terms of rate of exclusive breastfeeding, skin-to-skin contact, spousal support, partner violence…? It would be interesting to see whether cultural background differences contributed to factors related to breastfeeding.

4.      Breastfeeding at 1-4 weeks, 1-4 weeks appeared to be wide. The use of the wide time interval should be added to the limitations and be discussed. For example, a woman who breastfed at 1 week could stop by the 4th week. You could have misclassification bias.

5.      How did the results among CALD comparing to native mothers? It would be interesting to see related discussions, besides breastfeeding rates. Do CALD mothers have higher depressive symptoms and spousal violence? Please add more implications for caring for the diverse group.

Author Response

Reviewer 3

Comments and Suggestions for Authors

This study investigates the determinants of EBF cessation in the early postnatal period among CALD Australian mothers in Sydney. The manuscript is generally well written. The main concern with the study was the use of CALD as one group. It will be interesting to show sub-group results and provide more insights for cultural care.

Response

Thank you for the comment. As noted in the introduction section of the original manuscript, CALD is an Australian Government phrase that is often used for targeted interventions and research. Also, the small sample size in the present study limits our ability to performed detailed sub-group analyses as evident in the number of “no breastfeeding intention group”.

 1.      National data of proportion of CLAD among new mothers and that by regions/ethnicities could be presented in the Introduction. The proportion in the study could be compared to national data in order to shed light on representativeness of the sample.

Response

We agree with the reviewer that highlighting national data on CALD breastfeeding options is important. However, we noted in the original manuscript that quantitative data on breastfeeding among CALD mothers are limited in Australia; hence, one of the rationales for the study. Notably, additional text has been incorporated into the revised manuscript as suggested by the reviewer (Discussion section, Paragraph 6).

 2.      Rationale of selection of the potential predictors should be described. For example, do you speculate that CLAD mothers have lower social support or higher domestic violence? Do you include maternal work status as one of the predictors?

Response

The rationale for selecting the study variables was noted in the original manuscript (Study factors section): The exposure variables were broadly categorised into socio-demographic and health factors, which were selected based on previous studies [18, 19, 22, 23] and data availability.

 3.      CLAD Australian mothers are a very diverse group. Do women from different regions differ in terms of rate of exclusive breastfeeding, skin-to-skin contact, spousal support, partner violence…? It would be interesting to see whether cultural background differences contributed to factors related to breastfeeding.

Response

It was not unexpected that women would be different by breastfeeding prevalence given the global regions highlighted in Table 1.

 4.      Breastfeeding at 1-4 weeks, 1-4 weeks appeared to be wide. The use of the wide time interval should be added to the limitations and be discussed. For example, a woman who breastfed at 1 week could stop by the 4th week. You could have misclassification bias.

Response

We agree with the Reviewer. This limitation was noted in the original manuscript (Discussion section, Paragraph 7).

 5.      How did the results among CALD comparing to native mothers? It would be interesting to see related discussions, besides breastfeeding rates. Do CALD mothers have higher depressive symptoms and spousal violence? Please add more implications for caring for the diverse group.

Response

In response to Reviewer 1, a manuscript comparing breastfeeding patterns and depressive symptoms among CALD and non-CALD women based on a 5-year MCH data is been developed. We note that a discussion outside breastfeeding would be out of scope with the aim of the study. In response to reviewer 1 and 2, we have incorporated additional text into the Discussion section on how CALD mothers could be better supported to improve breastfeeding outcomes. Information relating to higher depressive symptoms are published elsewhere (Ogbo et al, 2018 & Eastwood et al., 2017).

References

·         Ogbo FA, Eastwood J, Hendry A, Jalaludin B, Agho KE, Barnett B, et al. Determinants of antenatal depression and postnatal depression in Australia. BMC Psychiatry. 2018;18(1):49.

·         Eastwood J, Ogbo FA, Hendry A, Noble J, Page A, Early Years Research Group. The impact of antenatal depression on perinatal outcomes in Australian women. Plos One. 2017;12(1):e0169907

Also, a manuscript that considers intimate partner violence is potentially acceptable for publication in BMC Pregnancy and Childbirth.